# Anti-Islet Autoantibodies in Type 1 Diabetes

**DOI:** 10.3390/ijms241210012

**Published:** 2023-06-11

**Authors:** Eiji Kawasaki

**Affiliations:** Diabetes Center, Shin-Koga Hospital, Kurume 830-8577, Japan; e-kawasaki@tenjinkai.or.jp; Tel.: +81-942-38-2222

**Keywords:** enzyme-linked immunosorbent assay, epitope, glutamic acid decarboxylase, latent-autoimmune diabetes in adults, prediction, type 1 diabetes

## Abstract

Anti-islet autoantibodies serve as key markers in immune-mediated type 1 diabetes (T1D) and slowly progressive T1D (SPIDDM), also known as latent autoimmune diabetes in adults (LADA). Autoantibodies to insulin (IAA), glutamic acid decarboxylase (GADA), tyrosine phosphatase-like protein IA-2 (IA-2A), and zinc transporter 8 (ZnT8A) are currently employed in the diagnosis, pathological analysis, and prediction of T1D. GADA can also be detected in non-diabetic patients with autoimmune diseases other than T1D and may not necessarily reflect insulitis. Conversely, IA-2A and ZnT8A serve as surrogate markers of pancreatic β-cell destruction. A combinatorial analysis of these four anti-islet autoantibodies demonstrated that 93–96% of acute-onset T1D and SPIDDM cases were diagnosed as immune-mediated T1D, while the majority of fulminant T1D cases were autoantibody-negative. Evaluating the epitopes and immunoglobulin subclasses of anti-islet autoantibodies help distinguish between diabetes-associated and non-diabetes-associated autoantibodies and is valuable for predicting future insulin deficiency in SPIDDM (LADA) patients. Additionally, GADA in T1D patients with autoimmune thyroid disease reveals the polyclonal expansion of autoantibody epitopes and immunoglobulin subclasses. Recent advancements in anti-islet autoantibody assays include nonradioactive fluid-phase assays and the simultaneous determination of multiple biochemically defined autoantibodies. Developing a high-throughput assay for detecting epitope-specific or immunoglobulin isotype-specific autoantibodies will facilitate a more accurate diagnosis and prediction of autoimmune disorders. The aim of this review is to summarize what is known about the clinical significance of anti-islet autoantibodies in the pathogenesis and diagnosis of T1D.

## 1. Introduction

Diabetes mellitus is a chronic metabolic disorder characterized by hyperglycemia, and is classified into four categories: type 1 diabetes (T1D), type 2 diabetes (T2D), specific types of diabetes due to other causes, and gestational diabetes [1]. Patients with untreated or uncontrolled hyperglycemia over a prolonged period of time may develop microvascular and macrovascular complications, such as diabetic neuropathy, retinopathy, nephropathy, cardiovascular and cerebrovascular disease, and peripheral vascular disease. Consequently, early detection or prediction of diabetes onset and timely intervention with appropriate medications, alongside medical nutrition therapy and exercise, are crucial.

T1D is an organ-specific autoimmune disease characterized by pancreatic β-cell destruction, leading to absolute insulin deficiency. Evidence supporting the autoimmune basis of T1D includes: (i) the presence of lymphocytic infiltration around and into the islets (termed insulitis), (ii) the appearance of autoantibodies to multiple islet autoantigens, (iii) the presence of both major histocompatibility complex (MHC)-linked and non-MHC-linked disease susceptibility genes, and (iv) the increased propensity to develop multiple organ-specific autoimmune diseases [2].

The risk of developing T1D varies considerably based on the country of residence and ethnicity, with Japan having one of the lowest incidence rates of T1D worldwide [3]. This variation may be attributed to differences in genetic background and environmental factors. In the current etiological classification of diabetes, T1D is divided into immune-mediated and idiopathic types, distinguished solely by the presence or absence of anti-islet autoantibodies in peripheral blood [1]. Furthermore, based on the rate of β-cell destruction, there are three T1D subtypes: fulminant T1D, acute-onset T1D, and slowly progressive T1D (SPIDDM), also known as latent autoimmune diabetes in adults (LADA) [4]. Since anti-islet autoantibodies are known to appear before disease onset, they serve as important humoral immune markers for predicting and diagnosing T1D. The aim of this review is to describe what is known regarding the clinical significance of anti-islet autoantibodies in the pathogenesis and diagnosis of T1D, and our recent findings on the prediction of future insulin deficiency in patients with SPIDDM (LADA) are highlighted.

## 2. History of Anti-Islet Autoantibody Discovery

Discovery of islet cell antibodies (ICA) as the first anti-islet autoantibodies in T1D was made by Bottazzo and coworkers in 1974 [5]. ICA detection involved using indirect immunofluorescence on the frozen pancreatic tissue sections of human blood group O, which may recognize various autoantigens. In 1982, Baekkeskov and coworkers discovered autoantibodies against an islet protein with a molecular weight of 64,000 (64 kDa antibody) using the immunoprecipitation method and ^35^S-methionine-labeled human islet cells [6]. Moreover, in 1983, Palmer and coworkers reported insulin autoantibody (IAA) in insulin naïve new-onset patients with T1D, measured by polyethylene glycol competitive assay using ^125^I-Tyr A14 human monoiodinated insulin [7].

To overcome the limitation of ICA assay, such as being time-consuming, requiring human pancreatic tissue, and yielding difficulty in obtaining quantitative results, extensive efforts have been made to identify target antigens against ICA using advanced molecular biological techniques such as molecular cloning, gel electrophoresis, polymerase chain reaction, and DNA microarray analysis. To date, more than 10 target antigens have been discovered (Table 1). After identifying the 64kDa islet protein as glutamic acid decarboxylase (GAD) in 1990 [8], several autoantibodies (Figure 1) were discovered. Currently, in addition to IAA and GAD autoantibodies (GADA), tyrosine phosphatase-like protein IA-2 autoantibodies (IA-2A) [9] and zinc transporter 8 autoantibodies (ZnT8A) [10] are employed for the diagnosis, pathological analysis, and prediction of T1D. Additionally, detection methods have evolved from immunohistochemical staining used in ICA to the radioimmunoassay (RIA), enzyme-linked immunosorbent assay (ELISA), and electrochemiluminescence (ECL) assay using recombinant autoantigens produced via prokaryotic and eukaryotic expression systems or in vitro transcription/translation system.

## 3. Pathophysiology of the Generation of Anti-Islet Autoantibodies

Although the role of B cells in the pathogenesis of T1D has been extensively studied in non-obese diabetic (NOD) mice, an animal model of human T1D, the mechanism of anti-islet autoantibody generation in humans remains largely unexplored and is currently unknown. However, it is assumed that genetic background, such as MHC class II gene, protein tyrosine phosphatase type 22 gene, and interleukin-2 receptor α gene, and environmental factors are associated with the anti-islet autoantibody generation [24]. In the NOD mice, it has been reported that the escape of islet-specific T cells into the periphery reduced regulatory T cell number and function, and the increased production of autoreactive B cells are thought to play key roles in anti-islet autoantibody generation [24,25]. Pinto and coworkers reported that increased numbers of thymic B cells and the formation of thymic germinal centers lead to a substantial increase in intrathymic autoantibody levels, resulting in the loss of certain medullary thymic epithelial cells, a decreased negative selection of autoreactive T cells, and an enhanced survival of insulin-reactive thymocytes [26].

## 4. Localization and Function of Autoantigens against Anti-Islet Autoantibodies

(1)Insulin

Insulin is a peptide hormone converted from proinsulin in pancreatic β-cells. Proinsulin consists of an A-chain (21 amino acids), a B-chain (30 amino acids), and a C-peptide (31 amino acids), and mature insulin is generated after C-peptide is excised by a series of proteolytic cleavages. The function of insulin is to regulate glucose levels in the blood and induce glucose storage in the liver, muscles, and adipose tissue.

(2)GAD 65 and GAD67

GAD is a rate-limiting enzyme that catalyzes a decarboxylation reaction to produce the neurotransmitter γ-aminobutyric acid from L-glutamate. There are two isoforms of GAD, GAD65 (585 amino acids), and GAD67 (594 amino acids), both of which are abundant in the central nervous system and pancreatic islets. In pancreatic β-cells, GAD65 and GAD67 are localized in the synaptic-like vesicles and cytosol, respectively.

(3)IA-2 and IA-2β/phogrin

IA-2 (also known as ICA512) is a transmembrane glycoprotein of the protein tyrosine phosphatase (PTP) family that colocalizes with IA-2β/phogrin and is expressed in insulin-secretory granule membranes in pancreatic β-cells [9,13]. IA-2 consists of 979 amino acids and IA-2β/phogrin consists of 1015 amino acids. Both polypeptides regulate insulin secretory granule content and pancreatic β-cell growth.

(4)ZnT8

ZnT8, a member of the zinc transporter family, is specifically localized in the insulin-secretory granule membrane of pancreatic β-cells and consists of 369 amino acids. ZnT8 is responsible for transporting zinc influx from the cytoplasm into insulin granules, facilitating insulin hexamer formation [17,18]. Zinc ions transported into the insulin-secreting granule cavity by ZnT8 are used to stabilize insulin hexamers, and zinc ions ejected outside the cells during insulin secretion have an autocrine effect on β-cells, or alternatively, exert a paracrine action on α-cells.

(5)Carboxypeptidase H

Carboxypeptidase H is an enzyme which converts proinsulin into insulin and C-peptide by catalyzing the release of C-terminal arginine or lysine residues from polypeptides. This enzyme is localized in the insulin secretory granules and granule membrane of pancreatic β-cells.

(6)ICA69

Islet-cell autoantigen 69 (ICA69) is localized in the insulin secretory granule membrane and consists of 483 amino acids. This protein is involved in the signaling and maturation of dene-core vesicles.

(7)GM2-1 ganglioside

GM2-1 ganglioside (N-acetyl neuraminic acid-galactose-galactosamine-galactosamine-glucose-ceramide) is a pancreatic monosialoganglioside migrating between GM2 and GM1 standards which is localized in the secretory granules in β-cells and non-β-cells. However, the function of GM2-1 ganglioside is still unknown.

(8)Heat shock protein 60

Heat shock proteins were originally described as “cellular stress responders” for their role as chaperones. Heat shock protein 60 (HSP60) consists of 569 amino acids and is localized in the insulin secretory granules. This protein assists in the correct folding of partially folded polypeptides and the presentation of antigen to MHC molecules.

(9)GLUT2

Glucose transporter-2 (GLUT2) is a subclass of GLUTs and consists of 524 amino acids. GLUT2 is expressed mainly in pancreatic β-cells of the pancreas, liver, and kidney. This transporter is localized on the β-cell surface membrane and is responsible for the transport of glucose from blood into β-cells.

(10)Tetraspanin-7

Tetraspanin-7 is expressed in the insulin-containing granule membranes of pancreatic β-cells and glucagon-producing α-cells. This protein consists of 249 amino acids and is involved in the regulation of Ca^2+^-dependent insulin exocytosis.

(11)ICA12/SOX13

ICA12/SOX13 belongs to the class D subgroup of SOX transcription factors and is localized in the cytoplasm and nucleus in β-cells and non-β-cells. ICA12/SOX13 consists of 604 amino acids and contains a leucine zipper domain. The function of this transcription factor in the islets is still unknown.

## 5. Significance of Anti-Islet Autoantibodies in the Pathophysiology of T1D

The Juvenile Diabetes Research Foundation (JDRF), American Diabetes Association (ADA), and Endocrine Society recommend classifying the natural history of T1D into three stages [27] (Figure 2). In Stage 1, autoimmunity to pancreatic islet cells is induced by cytotoxic T cells, and the destruction of pancreatic β-cells (insulitis) begins, even though blood glucose levels remain within the normal range. Two or more IAA, GADA, IA-2A, and ZnT8A, emerge in this stage, with the five-year and ten-year risks of T1D development being approximately 44% and 70%, respectively, while the lifetime risk approaches 100% [28]. However, it is currently understood that anti-islet autoantibodies do not directly cause pancreatic β-cell destruction; they arise as the result of β-cell destruction mediated by T-cells. In Stage 2, similar to Stage 1, individuals test positive for two or more anti-islet autoantibodies, and β-cell destruction progresses, leading to glucose intolerance or dysglycemia. The five-year risk of T1D development at this stage is approximately 75%, and the lifetime risk approaches 100% [29]. Stage 3 is marked by the manifestations of typical clinical symptoms and signs of T1D, including polyuria, polydipsia, weight loss, general fatigue, diabetic ketoacidosis, and others. In this stage, the amount of residual pancreatic β-cells decreases to 20–30% of normal levels. Since anti-islet autoantibodies appear in the peripheral blood during Stage 1 and Stage 2, they are used as a tool for predicting the onset of T1D. The number of positive autoantibodies is more important for prediction than the specific positive autoantibodies [30].

Additionally, a study involving first-degree relatives of patients with T1D reported that IAA or GADA appeared first, followed by IA-2A and ZnT8A directly before the onset of diabetes [31]. We observed a similar sequential emergence of anti-islet autoantibodies in patients with T1D who developed the condition during interferon therapy [32]. Based on this evidence, IA-2A and ZnT8A are considered surrogate markers for pancreatic β-cell destruction. This is supported by a report demonstrating the epitope spreading of IA-2A during the preclinical stage of T1D [33].

Conversely, as shown in Table 2, GADA is detected not only in patients with acute-onset T1D and SPIDDM, but also in some patients with fulminant T1D and non-diabetic patients with polyglandular autoimmune syndrome, autoimmune thyroid disease (AITD), and stiff-person syndrome. Therefore, unlike IA-2A and ZnT8A, it is presumed that GADA may not be a specific marker for pancreatic β-cell destruction. In our case, where GADA became re-elevated more than 10 years after the onset of T1D, anti-thyroid autoantibodies turned positive directly before the re-elevation of GADA [34]. Therefore, it is important to note that GADA may be associated with thyroid autoimmunity rather than insulitis in some instances.

## 6. Role of Anti-Islet Autoantibodies in the Diagnosis of T1D

In this section, we discuss the key considerations when diagnosing T1D using anti-islet autoantibodies. First, because distinguishing IAA from insulin antibodies produced by exogenous insulin injection is challenging, we considered patients positive for IAA if antibodies to insulin were present in insulin-naïve patients or those within 2 weeks of initiating insulin therapy. Our study using sera obtained within 2 weeks after onset revealed that the prevalence of IAA, GADA, IA-2A, and ZnT8A in patients with acute-onset T1D was 58, 80, 66, and 63%, respectively (Figure 3A). Similarly, the prevalence of these autoantibodies in SPIDDM patients was 41, 83, 27, and 27%, respectively (Figure 3B). Thus, a combinatorial analysis of the four autoantibodies indicated that 96% of acute-onset T1D and 93% of SPIDDM were immune-mediated T1D, while approximately 5% were classified as idiopathic T1D [35]. These data are consistent with previous reports from studies conducted in countries with Caucasoid populations [10,36,37,38]. In contrast, the majority of patients with fulminant T1D were negative for anti-islet autoantibodies (Figure 3C) and are classified into idiopathic T1D, suggesting that the mechanism of β-cell destruction in fulminant T1D may differ from acute-onset T1D and SPIDDM.

In Japan, medical insurance guidelines instruct physicians to measure GADA first when T1D is suspected. However, about 10% of non-fulminant T1D patients are positive for other anti-islet autoantibodies without GADA. Therefore, measuring multiple autoantibodies is crucial for the accurate diagnosis of immune-mediated T1D.

## 7. Epitopes for Anti-Islet Autoantibodies and Their Clinical Relevance

### 7.1. Insulin Autoantibodies

Fewer epitope analysis studies have been conducted on IAA compared to those of GADA and IA-2A. In an earlier study, the significance of amino acid A13 of the A-chain for the binding of IAA in T1D was demonstrated [39]. Furthermore, it was revealed that conformational epitope spanning amino acid residues A8–A13 on the A-chain and B1–B3 on the B-chain was the major binding site for IAA and was disease-associated. Another epitope analysis using a recombinant Fab of the insulin-specific monoclonal antibody reported that the IAA epitope was located in the A-chain residues A8–A10 [40,41]. However, radioimmunoassay for IAA does not distinguish between epitopes on the insulin molecule bound by IAA from insulin antibodies induced by exogenous insulin injection [38]. Therefore, Devendra and coworkers used the random phage-displayed peptide library and serum obtained from an IAA-positive T1D patient and an insulin-treated insulin autoimmune syndrome patient to examine the difference in epitopes bound by both antibodies [42]. As a result, they identified the two phagotopes (phage that carry peptides that mimic epitopes), designated IAS-9 and IDD-10, which were able to discriminate between diabetes-associated and non-diabetes associated insulin antibodies. This suggests that phage display technology could potentially be exploited to develop an IAA-specific RIA.

### 7.2. GAD Autoantibodies

GADA is present in 70–80% of prediabetic relatives and new-onset patients with T1D [30,35]. The major antigenic region of GADA has been determined using truncated peptide or chimeric proteins of GAD65 and GAD67 to maintain the conformational structure, as previous studies reported that GADA in patients with T1D recognizes the conformational structure of the GAD molecule [43]. Moreover, it has been reported that disease-associated GADA is directed against GAD65, and humoral immune reactivity to GAD67 is likely to be cross-reactive to GAD65 [44,45]. Figure 4A shows a schematic representation of the GAD65 and the localization of GADA epitopes in T1D. Previous studies have demonstrated that GADA in T1D patients recognizes disease-specific GAD65 epitopes, located at the middle and C-terminal regions of GAD65 [46,47]. However, these studies may have also detected epitopes that cross-react with the GAD67 autoantibody. To detect GAD65 autoantibody-specific epitopes, we performed a competitive radioimmunoassay with recombinant GAD67 protein using a series of GAD65/GAD67 chimeric constructs, and identified the autoantibody epitopes in the N-terminal region (amino acids 1–244; N), the middle domain (amino acids 245–359; E1 and amino acids 360–442; E3), and the C-terminal region (amino acids 443–585; E2) of GAD65 [48].

Although GADA epitope specificities remain relatively stable after the clinical onset of T1D, it has been reported that in genetically predisposed subjects, GADA is initially generated against the middle and C-terminal regions of GAD65. Furthermore, the autoimmune response may undergo intramolecular epitope spreading toward epitopes on the N-terminus and further epitopes located in the middle [49].

### 7.3. IA-2 Autoantibodies

As shown in Figure 4B, this protein is 979 amino acids long and comprises of a luminal domain (amino acids 27–576), transmembrane domain (amino acids 577–600), and cytoplasmic domain (amino acids 601–979). Although the PTP core sequence is found at amino acids 907–917 in the cytoplasmic domain, the expressed recombinant protein does not exhibit protein tyrosine phosphatase activity. Studies evaluating the biological properties of IA-2 have demonstrated its role as an important regulator of dense core vesicle number as well as glucose-induced and basal insulin secretion [12,14].

IA-2A is present in 60–70% of prediabetic relatives and new-onset patients with T1D [30,33,50]. We and others have analyzed IA-2A epitopes recognized by diabetic sera using a series of IA-2 fragments or IA-2/phogrin chimeric proteins, and found that the major epitopes are localized in the cytoplasmic domain [51]. Approximately 95% of T1D patients and prediabetic relatives who are IA-2A positive recognize the PTP-like domain (amino acids 687–979), whereas only 5% of sera react with the luminal domain [51,52]. Furthermore, our binding and competition analysis using multiple IA-2/phogrin chimeric constructs demonstrated that a major unique epitope for IA-2A is localized to amino acids 762–887. A conformational epitope associated with the C-terminal 31 amino acids of IA-2 is recognized by one-third of sera, and a minor epitope is located on amino acids 601–762 of IA-2. Notably, intramolecular epitope spreading was found for relatives of T1D patients who later progressed to T1D. However, relatives who remained nondiabetic exhibited a decrease in the number of recognized epitopes. These studies are consistent with the hypothesis that IA-2 may be recognized as a consequence of β-cell destruction [33].

Another important epitope has been mapped in the juxta-membrane domain of IA-2 (amino acids 601–629; IA-2JM). Our data demonstrated that the age of disease onset in patients with only IA-2JMA was significantly higher than that in patients who reacted with the PTP-like domain, suggesting that the autoantibody recognition of IA-2 epitopes in autoimmune diabetes is associated with the age of disease onset, which may reflect the intensity of the β-cell destruction process [53].

### 7.4. ZnT8 Autoantibodies

In 2007, Sladek and coworkers identified four loci containing variants that confer T2D risk through a genome-wide association study, including a non-synonymous polymorphism in the ZnT8 gene (solute carrier family 30 member 8, *SLC30A8*), rs13266634 (C/T), which causes an R325W modification in the protein sequence [54]. In the same year, Hutton and coworkers discovered ZnT8 as a major autoantigen in T1D, and ZnT8A has been recognized as one of the four major anti-islet autoantibodies [10].

ZnT8A are present in 50–60% of prediabetic relatives and new-onset patients with T1D [10,33,35]. As shown in Figure 4C, ZnT8 is a 369 amino acid polytopic transmembrane protein with cytoplasmic N- and C-terminal tails. It has been reported that ZnT8A recognizes 101 amino acids localized in the cytoplasmic C-terminal region. In particular, the amino acid residue 325 (R325W) defined by the *SLC30A8* polymorphism is critical for humoral autoimmunity to this autoantigen, and binding of ZnT8A against two isotypes (ZnT8-325R, ZnT8-325W) depends on the patient’s *SLC30A8* genotype [55,56]. Consequently, heterozygotes with the CT genotype respond to both ZnT8-325R and ZnT8-325W, while CC and TT homozygotes respond exclusively to ZnT8-325R or ZnT8-325W, respectively. Thus, individuals respond to endogenous ZnT8 protein determined by their own genome; and therefore, the current ZnT8A assay uses a hybrid protein of two ZnT8 isotypes as antigens.

Furthermore, Wenzlau and coworkers identified that residues 332R, 333E, 336K, and 340K contribute to a conformational ZnT8A epitope independent of residue 325 by comparing human and mouse chimeric ZnT8 proteins [57], suggesting that this epitope may add to the diagnostic utility of measuring ZnT8A.

### 7.5. Other Anti-Islet Autoantibodies

Other anti-islet autoantibodies include autoantibodies against GM2-1 ganglioside, HSP60, GLUT2, tetraspanin-7, and ICA12/SOX13 (Table 1). Among these, the epitopes of GM2-1 autoantibodies and GLUT2 autoantibodies have thus far not yet been analyzed. It has been reported that HSP60 autoantibodies recognized two epitope regions on HSP60 (amino acids 394–413 and amino acids 435–454). The first region is similar to the sequence found in GAD, whereas the second one overlaps with p277 T-cell epitope to a large extent [58]. Using a series of overlapping peptide fragments, Eugster and coworkers mapped autoepitopes recognized by tetraspsnin-7 autoantibodies and found that autoantibody epitopes lie predominantly within the first and third cytoplasmic domains of the protein. Further characterization of autoantibody binding to mutated constructs revealed that epitopes lie within a relatively short (20 amino acids) region represented by at least two of the three cytoplasmic domains, providing further evidence of the importance of protein conformation in antibody binding [59]. Furthermore, epitope mapping of ICA12/SOX13 autoantibodies using several truncated fragments of SOX13 suggests that autoantibodies are directed to at least two epitopes, one that requires amino acids 66–604, and a second confined within amino acids 327–604 [23].

## 8. Prediction of Future Insulin Deficiency in Patients with SPIDDM (LADA)

SPIDDM (also known as LADA) is characterized by the presence of anti-islet autoantibodies and a gradual decline in insulin secretory capacity. The Immunology of Diabetes Society defined LADA as follows: (1) the onset of diabetes > 35 years, (2) positive test for at least one of the known anti-islet autoantibodies, and (3) the requirement of insulin treatment > 6 months after the diagnosis of diabetes [60]. LADA encompasses anti-islet autoantibody-positive diabetic patients in both insulin-dependent and non-insulin-dependent states, which is nearly identical to SPIDDM. According to the recently revised diagnostic criteria for SPIDDM, the interval from diabetes diagnosis to the requirement of insulin treatment is >3 months [61]. Additionally, patients with exhausted endogenous insulin secretion (fasting C-peptide < 0.6 ng/mL) at the last observed time point are defined as “SPIDDM (definite)”. In contrast, anti-islet autoantibody-positive patients in a non-insulin-dependent state are classified as “SPIDDM (probable)” (Table 3). Using this diagnostic criterion, measuring anti-islet autoantibodies other than GADA results in an approximately 3-fold increase in the incidence of SPIDDM among non-insulin-treated diabetic patients compared with measuring GADA alone (2.0–2.4% vs. 7–8%) [62,63,64]. Indeed, in the Nagasaki Autoimmune Diabetes Intervention/Prevention Study, the prevalence of anti-islet autoantibodies other than GADA in insulin-naïve adult-onset diabetes was 8.6%, which is 2.6-fold compared to that of GADA (3.3%) (Figure 5). Since this subtype of T1D is generally indistinguishable from T2D at the time of diagnosis, measuring anti-islet autoantibodies is crucial for the early diagnosis and appropriate treatment of SPIDDM (LADA).

Anti-islet autoantibody positivity, especially ICA and GADA, is predictive for progression to a future insulin-dependent state after the diagnosis of diabetes. For example, the UKPDS (United Kingdom Prospective Diabetes Study) found that at least 50% of LADA patients required insulin treatment 6 years post-diagnosis [65]. However, not all SPIDDM (LADA) patients required insulin treatment, even after 10 years from diagnosis.

According to a nationwide survey [66] conducted by the Japan Diabetes Society, the predictors of progression to an insulin-dependent state include: (1) the age of onset is ≤47 years, (2) a period until GADA positive detection of ≤5 years, (3) GADA titer (RIA method) ≥13.6 U/mL, and (4) fasting C-peptide ≤ 0.65 ng/mL. Additionally, the number of positive anti-islet autoantibodies and GADA epitope recognition are also important for prediction. To identify the predictive markers for early insulin requirement in non-insulin-dependent SPIDDM (probable), we evaluated IAA, IA-2A, and ZnT8A along with GADA-specific epitope recognition in 47 GADA-positive diabetic patients [63]. Among these patients, 38% had one or more of IAA, IA-2A, or ZnT8A and 15% had two or more of these autoantibodies. A high GADA titer (≥10 U/mL), the presence of GADA-E1, and the presence of one or more among IAA, IA-2A, or ZnT8A at diagnosis marked the risk for early insulin therapy requirement. Furthermore, multiple anti-islet autoantibodies were the most relevant risk factors for the insulin requirement (odds ratio 13.77; 95% CI 2.77-68.45; *p* < 0.001) in a multivariate logistic regression analysis. Therefore, measuring anti-islet autoantibodies other than GADA and ICA is essential for predicting the progression risk of SPIDDM (LADA) patients.

## 9. Type 1 Diabetes and Associated Autoimmune Diseases

Clinicians should be aware that complications arising from additional autoimmune disorders are more frequently observed in T1D patients [67], with the most common organ-specific autoimmune disease associated with T1D being autoimmune thyroid disease (AITD), such as Graves’ disease and Hashimoto’s thyroiditis, affecting more than 90% of people with T1D and autoimmune disorders [68,69]. Additionally, it has been reported that children, particularly girls, with T1D are at an increased risk for developing other autoimmune diseases, with the prevalence of anti-thyroid autoantibodies at disease onset being about 20% [70]. Furthermore, the prevalence of anti-thyroid antibodies increases with age, and the presence of these antibodies at diagnosis of T1D is predictive of future thyroid disease [70]. Long-term follow-up suggests that up to 30% of patients with T1D develop AITD [71], and patients with thyroid autoimmunity are 18 times more likely to develop AITD than those without [72]. Therefore, to enable an early diagnosis of AITD in children with T1D, the International Society for Pediatric and Adolescent Diabetes Consensus Clinical Guidelines recommends screening for thyroid function by analyzing circulating TSH at the time of diabetes diagnosis and every second year thereafter in asymptomatic individuals without goiter, and with more frequent screening in patients with goiter present [73].

To characterize the T1D patients with coexisting AITD (referred to as autoimmune polyendocrine syndrome type 3 variant, APS3v), we analyzed their clinical characteristics compared to those without AITD [74]. Patients with APS3v demonstrated a significant female predominance, a slower and older age of onset for T1D, and a higher prevalence and level of GADA. Furthermore, among patients with Graves’ disease, 60% developed the disease before T1D, 30% had antecedent T1D, and 10% developed T1D and Graves’ disease simultaneously. The interval between the onset of T1D and Graves’ disease was <10 years in most cases, but approached or exceeded 20 years in some instances.

The prevalence of GADA in nondiabetic patients with ATID is 8–10% [75,76,77], and it has been reported that GADA positivity is associated with a decreased insulin secretion capacity [78], suggesting that GADA positivity could be a marker of subclinical insulitis. Furthermore, the incidence of T1D/SPIDDM in anti-islet autoantibody-positive patients with Graves’ disease has been reported to be 2.5 times higher than in autoantibody-negative patients [79]. The exact reasons why the level of GADA in T1D patients with AITD is extremely high compared to patients without AITD remain unclear. Previous studies have reported that GAD65 is expressed in the thyroid gland [80,81], and T1D patients with AITD demonstrated a higher polyclonality in GADA epitope recognition and IgG subclasses compared to patients without AITD (Figure 6). Forty-eight percent of T1D patients with AITD recognized GAD65 N-terminal region, middle domain, and C-terminal region epitopes, while the same was true for only 9% of those without AITD (*p* < 0.005). Additionally, the prevalence of patients with two or more IgG subclasses was significantly higher in T1D patients with AITD than those without AITD (57% vs. 17%, *p* < 0.001). Therefore, the overproduction of GADA in T1D patients with AITD might be attributable to the activation of polyclonal B-lymphocyte response by GAD in the thyroid gland [75].

## 10. Recent Advances in Anti-Islet Autoantibody Assay

As described in the previous section, ICA is detected by immunohistochemistry using indirect immunofluorescence or immunoenzymatic techniques on the frozen sections of human blood group O pancreas. Achieving a concordance of results obtained from different laboratories is essential for comparing studies from various centers. To address this issue, a series of international workshops on the standardization of ICA was initiated in 1985 under the auspices of the Immunology and Diabetes Workshop (currently Immunology and Diabetes Society). However, wide variations in results persisted despite using the same experimental protocol [82]. Therefore, subsequent workshops employed standard curves constructed from a reference serum (JDRF standard), which led to a decrease in interlaboratory variation [83].

Following the discovery of GAD as a major target antigen against ICA, numerous recombinant anti-islet autoantibody assays, including radioligand binding assays (RBA), RIA, and ELISA, have been developed. With these assays, the reported prevalence of anti-islet autoantibodies such as GADA in patients with recently diagnosed T1D ranged widely from 25% [84] to 80% [85,86]. To standardize the recombinant anti-islet autoantibody assays, the first GADA workshop was held in 1993. In contrast to the ICA workshop, considerable concordance in ranking GADA levels in 16 different samples was observed among various assay formats [87]. In the second GADA workshop, which used a large number of T1D and control sera, the RBA format demonstrated greater sensitivity than conventional ELISA [88]. Solid-phase ELISA formats can detect insulin-binding antibodies in patients receiving insulin injections. However, these assays fail to detect IAA associated with disease risks, likely due to the high apparent affinity (10^10^) and extremely low capacity (10^−12^) of prediabetes autoantibodies [39]. Furthermore, the widely used fluid-phase RBA has proven to be challenging for many laboratories to implement. In response to the need for improved anti-islet autoantibody assays, a nonradioactive fluid-phase assay has been developed that offers high sensitivity and specificity. This assay employs a modified ELISA format, which is based on the autoantibodies’ ability to form a bridge between recombinant autoantigens (such as GAD, IA-2, and ZnT8) coated on the ELISA plate and biotin-labeled corresponding autoantigens, allowing for the detection of autoantibody-bound antigens rather than immunoglobulins themselves [89]. Using the same assay format, an anti-islet autoantibody assay has recently been developed that allows for the simultaneous measurement of multiple autoantibodies in a single well (3 Screen ICA ELISA) [90]. Although this assay cannot distinguish which of the three autoantibodies are present, it may serve as a useful screening test for T1D and facilitate the efficient diagnosis of immune-mediated T1D. Other sensitive high-throughput assays effective for large-scale screening include a multiplex ECL assay and a multiplex agglutination-PCR autoantibody assay that combines all four biochemically defined anti-islet autoantibodies [91,92]. These innovations from around the globe will enhance our ability to identify high-risk individuals for T1D more accurately and efficiently at an early stage, promoting the advancement of early interventions for the benefit of public health.

## 11. Anti-Islet Autoantibodies in Trials of Novel Therapeutic Approaches for the Preservation of β-Cell Function

The most important goal of diabetes management is to improve the quality of life and extend the lifespan by preventing micro- and macro-vascular diseases. To achieve this goal, it is crucial to maintain good glycemic control and residual β-cell function. Over the past 30 years, while numerous clinical trials regarding primary prevention have been conducted, to date, no treatment or agent has proven to be effective in preventing T1D in susceptible individuals. However, several therapeutic agents have demonstrated usefulness in delaying progression from Stage 2 to Stage 3 T1D while also preserving β-cell function in Stage 3. These agents include abatacept (anti-CTLA-4), alefacept (a fusion protein of soluble lymphocyte function antigen with Fc fragments of IgG1), rituximab (anti-CD20), and anti-thymocyte globulin, which preserve C-peptide production compared with placebo in Stage 3 T1D [93,94,95,96]. Moreover, teplizumab (anti-CD3) has been shown to delay progression from Stage 2 to Stage 3 T1D by a median of 3 years [97,98]. Based on these results, in November 2022, the U.S. Food and Drug Administration approved teplizumab as the first drug to delay progression from Stage 2 to Stage 3 T1D in adults and children > 8 years. The previous prevention trials targeted the subjects who were positive for at least one or more anti-islet autoantibodies. However, since these studies used the RBA method for measuring anti-islet autoantibodies, there is a possibility that low-risk subjects were also included. Therefore, it is crucial to screen subjects with assays that can exclusively detect high-affinity autoantibodies in order to verify more reliable preventive effects. Additionally, the ability to assess multiple autoantibodies in a single test should prove valuable for future interventional trials.

## 12. Conclusions

This article focused on reviewing the current understanding of anti-islet autoantibodies in T1D. The clinical utilities of anti-islet autoantibodies in patients with diabetes include diagnosis (immune-mediated or idiopathic), prediction (progressor or non-progressor), and the understanding of pathophysiology (insulitis-specific or nonspecific phenomenon) (Figure 7). Since the autoantibody level of anti-islet autoantibodies decreases with disease duration and can become negative, it is essential to measure them early in the onset of T1D for accurate diagnosis. SPIDDM or LADA is often indistinguishable from T2D; therefore, an earlier measurement of anti-islet autoantibodies is of great clinical importance for early diagnosis and appropriate treatment. In addition to the anti-islet autoantibody profiles, the age of onset and genetic risk score should also be considered for risk triage. Furthermore, the development of a high-throughput assay to detect epitope-specific or immunoglobulin isotype-specific autoantibodies should warrant the accurate diagnosis and prediction of autoimmune disorders. Furthermore, the new type of autoantibody assays, which can simultaneously measure multiple autoantibodies, have the advantages of high sensitivity and specificity, and the ability to measure a large number of samples, making it suitable for a large-scale population screening of T1D.

## Figures and Tables

**Figure 1 ijms-24-10012-f001:**
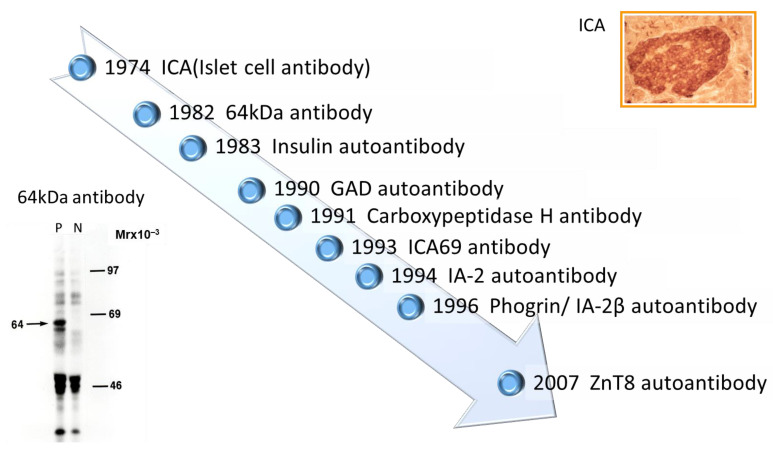
Chronology of anti-islet autoantibody discovery. Anti-islet autoantibodies used for prediction and diagnoses of T1D are IAA, GADA, IA-2A, and ZnT8A.

**Figure 2 ijms-24-10012-f002:**
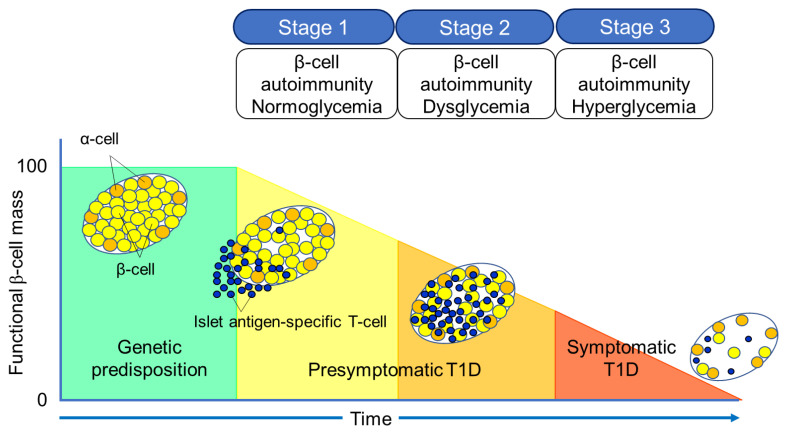
Three stages of natural history of T1D.

**Figure 3 ijms-24-10012-f003:**
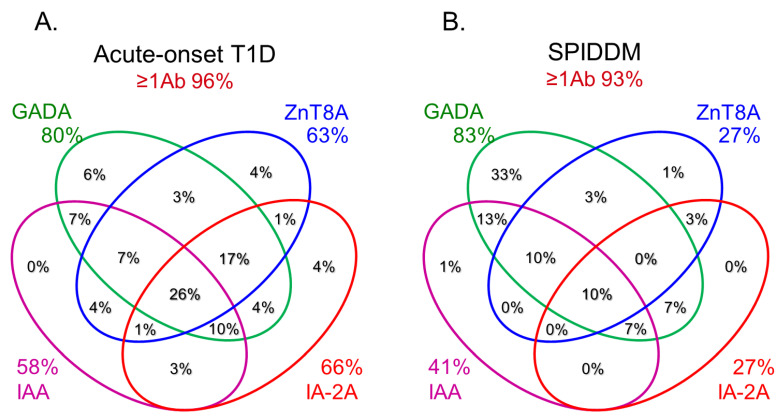
Combinatorial analysis of anti-islet autoantibodies in patients with acute-onset T1D (**A**), SPIDDM (**B**), and fulminant T1D (**C**) Adapted with permission from Ref. [35].

**Figure 4 ijms-24-10012-f004:**
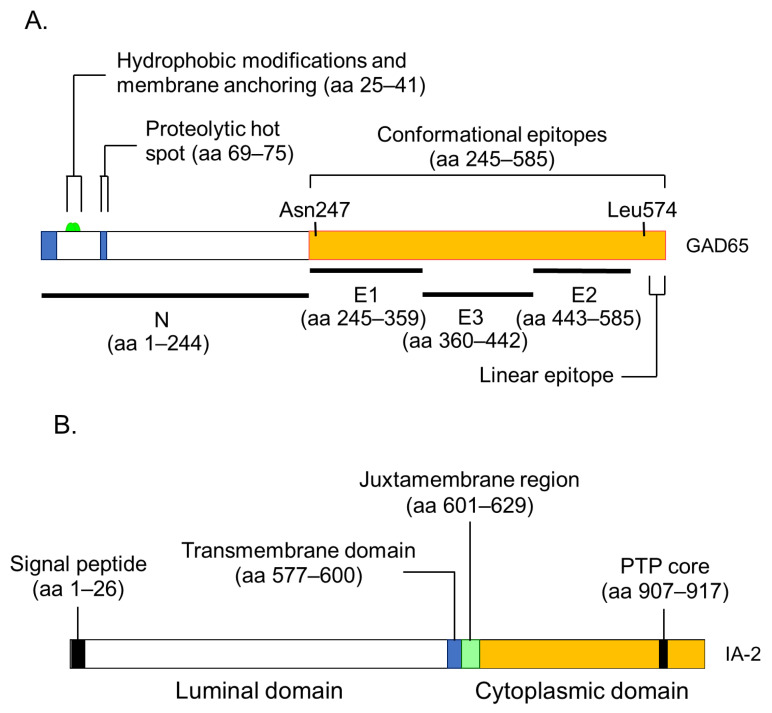
Illustration of antigenic epitopes recognized by T1D sera in GAD65 (**A**), IA-2 (**B**), and ZnT8 (**C**) proteins.

**Figure 5 ijms-24-10012-f005:**
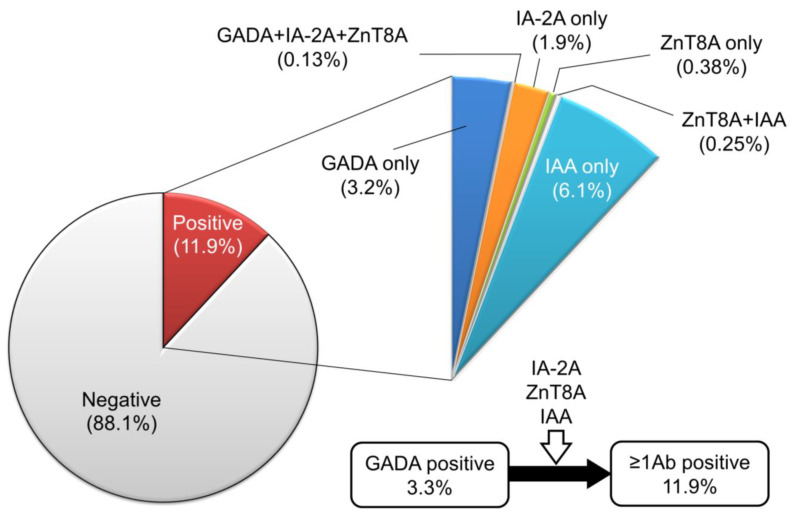
Prevalence of GADA, IA-2A, ZnT8A, and IAA in 788 insulin-naïve adult-onset patients with diabetes.

**Figure 6 ijms-24-10012-f006:**
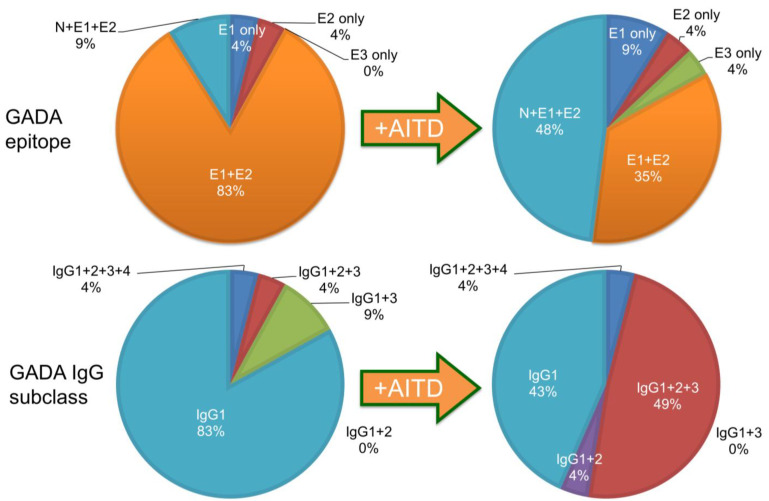
Comparison of GADA epitopes (**top**) and IgG subclasses (**bottom**) in T1D patients without and with autoimmune thyroid disease. (**Left panel**) T1D without AITD. (**Right panel**) T1D with AITD.

**Figure 7 ijms-24-10012-f007:**
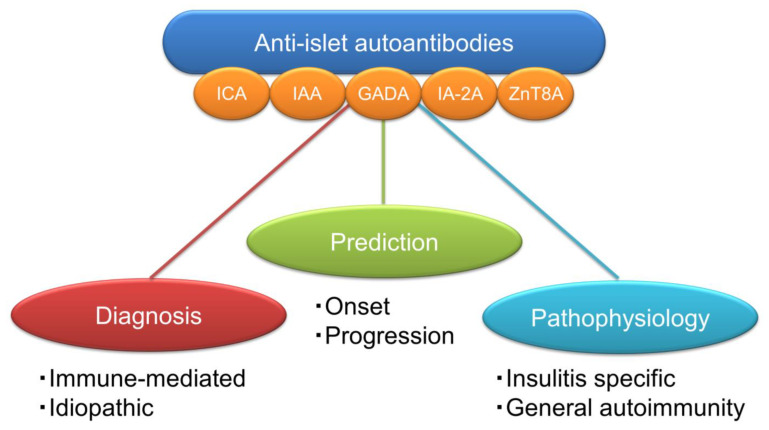
Clinical utilities of anti-islet autoantibodies in patients with diabetes.

**Table 1 ijms-24-10012-t001:** Localization and function of autoantigens against anti-islet autoantibodies.

Name of Antigen	Localization	Function	Reference
Insulin	Insulin secretory granules	Regulate glucose levels in the blood and induce glucose storage in the liver, muscles, and adipose tissue	[7]
GAD65	Synaptic-like vesicles in the cytoplasm of β-cells	Rate-limiting enzyme engaged in the synthesis of the neurotransmitter γ-aminobutyric acid from L-glutamate	[8]
GAD67	Cytosol of β-cells	Rate-limiting enzyme engaged in the synthesis of the neurotransmitter γ-aminobutyric acid from L-glutamate	[11]
IA-2	Insulin secretory granule membrane	Regulate insulin secretory granule content and β-cell growth	[9,12]
Phogrin/IA-2β	Insulin secretory granule membrane	Regulate insulin secretory granule content and β-cell growth	[13,14]
Carboxypeptidase H	Insulin secretory granules and granule membrane	Convert proinsulin into insulin and C-peptide by catalyzing the release of C-terminal arginine or lysine residues from polypeptides	[15]
ICA69	Insulin secretory granule membrane	Dense-core vesicles signaling and maturation	[16]
ZnT8	Insulin secretory granule membrane	Transport zinc ion from the cytosol into the insulin secretory granules	[17,18]
GM2-1 ganglioside	Secretory granules in β-cells and non-β-cells	unknown	[19]
Heat shock protein 60	Insulin secretory granules	Assist correct folding of partially folded polypeptides and presentation of antigen to MHC molecules	[20]
GLUT2	β-cell surface membrane	Uptake glucose from the blood into β-cells	[21]
Tetraspanin-7	Insulin secretory granule membrane	Regulate Ca^2+^-dependent insulin exocytosis	[22]
ICA12/SOX13	Cytoplasm and nucleus in β-cells and non-β-cells	Transcription factor (Function in the islets is unknown)	[23]

**Table 2 ijms-24-10012-t002:** Disease specificity of GAD autoantibodies.

Subject	Prevalence
Healthy control	<1%
Acute-onset type 1 diabetes (at onset)	60–80%
Fulminant type 1 diabetes	5–9%
LADA (SPIDDM)	100%
Type 2 diabetes (diet/OHA)	4–5%
Polyglandular autoimmune syndrome, type 1	30–40%
Polyglandular autoimmune syndrome, type 2	30–50%
Autoimmune thyroid disease	6–8%
Stiff-person syndrome	60–70%

LADA, latent-autoimmune diabetes in adults; SPIDDM, slowly progressive insulin-dependent diabetes; OHA, oral hypoglycemic agents.

**Table 3 ijms-24-10012-t003:** Diagnostic criteria (2023) of slowly progressive type 1 diabetes (SPIDDM).

**Required Item:**
(1) The presence of anti-islet autoantibodies at some time point during the disease course ^a^;
(2) The absence of ketosis or ketoacidosis at the diagnosis of diabetes and the unnecessity for insulin treatment to correct hyperglycemia immediately after diagnosis in principle;
(3) The gradual decrease in insulin secretion overtime, requirement of insulin treatment more than 3 months ^b^ after diagnosis of diabetes, and exhausted endogenous insulin secretion (fasting serum C-peptide immunoreactivity < 0.6 ng/mL) at last observed time point.
**Judgement:**
When the case fulfills the criteria of all of the three described above ((1), (2), and (3)), the case is diagnosed with “slowly progressive type 1 diabetes (definite)”; when the case fulfills the criteria only (1) and (2), but not (3), the case is diagnosed with “slowly progressive type 1 diabetes (probable)”.

^a^ Anti-islet autoantibodies include glutamic acid decarboxylase (GAD) autoantibody, insulinoma-associated antigen-2 (IA-2) autoantibody, islet cell antibody (ICA), zinc transporter 8 (ZnT8) autoantibody or insulin autoantibody (IAA). The measurement of IAA should be performed before starting insulin treatment. ^b^ More than 6 months in a typical case.

## Data Availability

Not applicable.

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
