# Peer review of "Anti-Islet Autoantibodies in Type 1 Diabetes"

_ijms, 2023, doi:10.3390/ijms241210012_

Round 1

Reviewer 1 Report

This paper from my point of view is not completaly original one

The key, the question is: with is your contibution to this fact?

Ypu can fiend a lot of papers about

https://pubmed.ncbi.nlm.nih.gov/?term=antibodies+and+dm+type+1

-What is the main question addressed by the research? Anti-Islet Autoantibodies in Type 1 Diabetes. But the ask is which is new in this fact? -Is it relevant and interesting? yes -How original is the topic? Is not -What does it add to the subject area compared with other published material? There are a lot papers very similar. In this paper which is the new one? -Is the paper well written? Yes -Is the text clear and easy to read? Yes is easy and clear. Please clarifficy a litte more objetives -Are the conclusions consistent with the evidence and arguments presented? Yes all of them are presented. But i miss whta happnes with new type antibodies. Why peaple have not antibodies
Plase redisign the article

Author Response

I thank you for your evaluation of our manuscript.  I have revised the manuscript according to your comments.

Comments and Suggestions for Authors

This paper from my point of view is not completaly original one

Response: This paper was written as an original review article by myself.  The submitted manuscript has not been published previously, nor is under consideration for publication elsewhere.  Therefore, this review article is completely original one.

The key, the question is: with is your contibution to this fact?

Response: I have selected the topics to be covered in this review article, and have compiled a number of previously reported data by topic.

Ypu can fiend a lot of papers about

https://pubmed.ncbi.nlm.nih.gov/?term=antibodies+and+dm+type+1

-What is the main question addressed by the research? Anti-Islet Autoantibodies in Type 1 Diabetes. But the ask is which is new in this fact?

Response: The aim of this review article is to describe what is known regarding the clinical significance of anti-islet autoantibodies in the pathogenesis and diagnosis of type 1 diabetes, and our recent findings on the prediction of future insulin deficiency in patients with SPIDDM (LADA) are highlighted.

-Is it relevant and interesting? yes

-How original is the topic? Is not

-What does it add to the subject area compared with other published material? There are a lot papers very similar. In this paper which is the new one?

Response: As I made response to your first comment, this review article is original one.  To the best of my knowledge, there are no review papers describing the relationship between anti-islet autoantibodies and the pathophysiology of type 1 diabetes, the revised diagnostic criteria for Japanese SPIDDM, and predictive markers for the progression to insulin-deficient state in patients with SPIDDM.

-Is the paper well written? Yes

-Is the text clear and easy to read? Yes is easy and clear. Please clarifficy a litte more objetives

Response: I added the following sentence as the objective of this review article.

“The aim of this review is to describe what is known regarding the clinical significance of anti-islet autoantibodies in the pathogenesis and diagnosis of type 1 diabetes, and our recent findings on the prediction of future insulin deficiency in patients with SPIDDM (LADA) are highlighted.”

-Are the conclusions consistent with the evidence and arguments presented? Yes all of them are presented. But i miss whta happnes with new type antibodies. Why peaple have not antibodies

Plase redisign the article

Response: The advantages of the new type of autoantibody assay, which can simultaneously measure multiple autoantibodies, are its high sensitivity and specificity, and its ability to measure a large number of samples, making it suitable for screening in the general population.  The following sentence was added in the “Conclusions” section.

“Besides, the new type of autoantibody assays, which can simultaneously measure multiple autoantibodies, have the advantages of high sensitivity and specificity, and the ability to measure a large number of samples, making it suitable for large-scale population screening of T1D.”

Although the role of B cells in the pathogenesis of type 1 diabetes has been extensively studied in non-obese diabetic mice, an animal model of human type 1 diabetes, the mechanism of anti-islet autoantibody generation in humans remains largely unexplored and is currently unknown.  Therefore, the question of why people do not have antibodies should be analyzed in the future.

Reviewer 2 Report

It is a very interesting and well presented review of the current literature. However there few things need improvement. 

Authors presented very well the current literature for the antibodies, although the do not refer anywhere in their study about the aim.

In addittion, discussion is missing.

Finally, in their conclusion they refer to the novel therapeutic approaches for reducing antibodies. However it could be better if they write a section in the main part of their manuscript in order to explain better the link of this review with these agents

English language is good 

Author Response

I thank you for your evaluation of our manuscript.  I have revised the manuscript according to your comments.

Comments and Suggestions for Authors

It is a very interesting and well-presented review of the current literature. However there few things need improvement.

Authors presented very well the current literature for the antibodies, although the do not refer anywhere in their study about the aim.

Response: Thank you for your comment.  In the “Introduction” section, the sentence “The aim of this review is to describe what is known regarding the clinical significance of anti-islet autoantibodies in the pathogenesis and diagnosis of type 1 diabetes, and our recent findings on the prediction of future insulin deficiency in patients with SPIDDM (LADA) are highlighted.” was added.

In addittion, discussion is missing.

Response: It is not necessary to make the “Discussion” section for the review article.  In each section, several discussions are provided.

Finally, in their conclusion they refer to the novel therapeutic approaches for reducing antibodies. However, it could be better if they write a section in the main part of their manuscript in order to explain better the link of this review with these agents.

Response: According to your comment, I have created a new section called "Novel therapeutic approaches for the preservation of β-cell function".

Reviewer 3 Report

This manuscript is interesting for  the medicine community. The topic is original and actual. Current research is focused on Anti-Islet Autoantibodies in Type-1 Diabetes, which serve as key markers in type-1 diabetes.

A concise analysis of current developments regarding the significance and clinical utility of anti-islet autoantibodies has been provided in brief research.

Current and anticipated future challenges and opportunities in the sector are also highlighted.

The aim of the study is clear and the authors provided adequate information on how they conclude their results.

The introduction provides sufficient background. The results are clearly presented. The conclusions supported by the data. The manuscript good illustrated and interesting to read.

The references are relevant and generally recent and include appropriate studies.

Author Response

I thank you for your positive evaluation of our manuscript.  I have revised the manuscript according to other reviewers’ comments.

Comments and Suggestions for Authors

This manuscript is interesting for the medicine community. The topic is original and actual.

Current research is focused on Anti-Islet Autoantibodies in Type-1 Diabetes, which serve as key markers in type-1 diabetes.

A concise analysis of current developments regarding the significance and clinical utility of anti-islet autoantibodies has been provided in brief research.

Current and anticipated future challenges and opportunities in the sector are also highlighted.

The aim of the study is clear and the authors provided adequate information on how they conclude their results.

The introduction provides sufficient background. The results are clearly presented. The conclusions supported by the data. The manuscript good illustrated and interesting to read.

The references are relevant and generally recent and include appropriate studies.

Reviewer 4 Report

In this study by Kawasaki, a review of islet-specific auto-antibodies and their detection was performed. With the increasing prevalence of T1D, it has become very important to pay attention to autoantibody generation and detection as well as identify novel biomarkers for early detection. Overall, the article is well written; however, the author needs to address the following concerns for acceptance:

  1. The author needs to include a section on the physiology of the generation of autoantibodies. 
  2. Include critical functions of all the 10 islet proteins (antigens) against which autoantibodies are produced. Section 5 would be a good place to add this.
  3. The figure legends must be descriptive.
  4. Line 70: advanced molecular biological techniques: such as?
  5. Line 71: more than 10 target antigens have been discovered: List and functions (See concern no.2)
  6.  Line 79: Briefly explain recombinant autoantigens
  7. Figure 2: What do blue dots represent?
  8. Line 172: What are phagotopes?
  9. Minor errors include unnecessary spaces in the text.

Author Response

I thank you for your evaluation of our manuscript.  I have revised the manuscript according to your comments.

Comments and Suggestions for Authors

In this study by Kawasaki, a review of islet-specific auto-antibodies and their detection was performed. With the increasing prevalence of T1D, it has become very important to pay attention to autoantibody generation and detection as well as identify novel biomarkers for early detection. Overall, the article is well written; however, the author needs to address the following concerns for acceptance:

  1. The author needs to include a section on the physiology of the generation of autoantibodies.

Response: Although the role of B cells in the pathogenesis of type 1 diabetes has been extensively studied in non-obese diabetic mice, an animal model of human type 1 diabetes, the mechanism of anti-islet autoantibody generation in humans remains largely unexplored and is currently unknown.  Therefore, a section on the physiology of anti-islet autoantibody generation could not be included in this review article.

  1. Include critical functions of all the 10 islet proteins (antigens) against which autoantibodies are produced. Section 5 would be a good place to add this.

Response: I made a new Table 1 which described on the localization and functions of autoantigens against anti-islet autoantibodies.  In section 5, the descriptions regarding the functions of each autoantigen were added.

  1. The figure legends must be descriptive.

Response: According to your suggestion, the figure legends have been modified to be descriptive.

  1. Line 70: advanced molecular biological techniques: such as?

Response: According to your comment, the words “molecular cloning, gel electrophoresis, polymerase chain reaction, and DNA microarray analysis” were added.

  1. Line 71: more than 10 target antigens have been discovered: List and functions (See concern no.2)

Response: I made a new Table 1 which described on the localization and functions of autoantigens against anti-islet autoantibodies.

  1. Line 79: Briefly explain recombinant autoantigens

Response: The words “produced via prokaryotic and eukaryotic expression systems or in vitro transcription/translation system” were added.

  1. Figure 2: What do blue dots represent?

Response: Blue dots are islet antigen-specific T cells infiltrating the islets.  I added description in the Figure 2.

  1. Line 172: What are phagotopes?

Response: “Phagotopes” are phage that carry peptides that mimic epitopes.  I added the explanation of phagotpes in the text.

  1. Minor errors include unnecessary spaces in the text.

Response: I thought the unnecessary spacing was due to full justification, so I changed the formatting to left justified.

Round 2

Reviewer 1 Report

Now yes We would accept this version

Author Response

Comments and Suggestions for Authors

Now yes We would accept this version

Response: I thank you for your positive evaluation of our manuscript.  I have revised the manuscript according to other reviewers’ comments.

Reviewer 2 Report

Authors made good effort and improved their manuscript.

However, in my opinion they should write a section with discussion in order to link the antibodies with novel therapeutic approaches.  

No comments

Author Response

I thank you for your re-evaluation of our manuscript.  I have revised the manuscript according to your comments.

Comments and Suggestions for Authors

Authors made good effort and improved their manuscript.

However, in my opinion they should write a section with discussion in order to link the antibodies with novel therapeutic approaches.

Response: According to your suggestion, I changed the title of this section from “Novel therapeutic approaches for the preservation of β-cell function” to “Anti-islet autoantibodies in trials of novel therapeutic approaches for the preservation of β-cell function” and added the following sentences as discussion.

“The previous prevention trials targeted the subjects who were positive for at least one or more anti-islet autoantibodies.  However, since these studies used the RBA method for measuring anti-islet autoantibodies, there is a possibility that low-risk subjects were also included.  Therefore, it is crucial to screen subjects with assays that can exclusively detect high-affinity autoantibodies in order to verify more reliable preventive effects.  Additionally, the ability to assess multiple autoantibodies in a single test should prove valuable for future interventional trials.”.

Reviewer 4 Report

The author has addressed most concerns however the the first two concerns still remain unanswered.

1. Although human auto-antibody production is not well explored, still author's perspectives are critical. Studies on NOD mice and other in vivo models are critical to be described in this scenario.

2. Finally the Table 1 description does not match author's rebuttal - it no where describes  "localization and functions of autoantigens against anti-islet autoantibodies". The author needs to correct it and add the correct details.

3. Section 5 does not include description on ALL 10 ISLET AUTOANTIBODIES. This needs to be corrected

Author Response

I thank you for your re-evaluation of our manuscript.  I have revised the manuscript according to your comments.

Comments and Suggestions for Authors

The author has addressed most concerns however the first two concerns still remain unanswered.

  1. Although human auto-antibody production is not well explored, still author's perspectives are critical. Studies on NOD mice and other in vivo models are critical to be described in this scenario.

Response: According to your comments, new section “Pathophysiology of the Generation of Anti-Islet Autoantibodies” was added in the re-revised manuscript.

  1. Finally the Table 1 description does not match author's rebuttal - it no where describes "localization and functions of autoantigens against anti-islet autoantibodies". The author needs to correct it and add the correct details.

Response: All islet autoantibodies listed in Table 1 were described in this section.

  1. Section 5 does not include description on ALL 10 ISLET AUTOANTIBODIES. This needs to be corrected.

Response: All islet autoantibodies listed in Table 1 were described in this section.

Round 3

Reviewer 4 Report

The author says that the comments are addressed but none of the comments have been addressed. Unless the comments are addressed the manuscript cannot be accepted.

1. Although human auto-antibody production is not well explored, still author's perspectives are critical. Studies on NOD mice and other in vivo models are critical to be described in this scenario. STILL NOT ADDRESSED

2. Finally the Table 1 description does not match author's rebuttal - it no where describes  "localization and functions of autoantigens against anti-islet autoantibodies". The author needs to correct it and add the correct details. THE TABLE 1 SUBMITTED BY THE AUTHOR IS SOMETHING ELSE

3. Section 5 does not include description on ALL 10 ISLET AUTOANTIBODIES. This needs to be corrected. AGAIN.. THE AUTHORS HAVE NOT ADDED THE DESCRIPTION OF ALL THE AUTOANTIBODIES IN SECTION 5.

Author Response

Please see in the attachment for the revised version

Round 4

Reviewer 4 Report

Thank you for submitting the correct version of the rebuttal. This file looks good and all the comments have been addressed satisfactorily. I would like to commend the authors for patiently addressing the concerns and positively willing improve the article.